# EAU Biochemical Recurrence Risk Classification and PSA Kinetics Have No Value for Patient Selection in PSMA-Radioguided Surgery (PSMA-RGS) for Oligorecurrent Prostate Cancer

**DOI:** 10.3390/cancers15205008

**Published:** 2023-10-16

**Authors:** Fabian Falkenbach, Francesca Ambrosini, Pierre Tennstedt, Matthias Eiber, Matthias M. Heck, Felix Preisser, Markus Graefen, Lars Budäus, Daniel Koehler, Sophie Knipper, Tobias Maurer

**Affiliations:** 1Martini-Klinik Prostate Cancer Center, University Medical Center Hamburg-Eppendorf, Martinistr. 52, 20246 Hamburg, Germany; 2IRCCS Ospedale Policlinico San Martino, 16132 Genoa, Italy; 3Department of Urology, Rechts der Isar Medical Center, Technical University of Munich, 81675 Munich, Germany; 4Department of Diagnostic and Interventional Radiology and Nuclear Medicine, University Medical Center Hamburg-Eppendorf, 20246 Hamburg, Germany; 5Department of Urology, Vivantes Klinikum am Urban, 10967 Berlin, Germany; 6Department of Urology, University Medical Center Hamburg-Eppendorf, 20246 Hamburg, Germany

**Keywords:** hormone-sensitive, PSMA, positron emission tomography, metastasis-directed therapy, oligorecurrent prostate cancer, radioguided surgery, PSA doubling time, PSA velocity, PSA kinetics

## Abstract

**Simple Summary:**

In prostate cancer patients with low-volume/oligo-metastatic relapse after radical prostatectomy, radioguided surgery against prostate-specific membrane antigen (PSMA-RGS) might be an option. However, patient selection remains difficult. In this study, we analyzed if the EAU risk stratification for biochemical recurrence (BCR) and the PSA development over time predict the oncological success of PSMA-RGS. We found that neither EAU BCR risk groups nor PSA doubling time and velocity are significant predictors for a complete biochemical response or prolonged biochemical recurrence- or subsequent therapy-free survival. Therefore, PSMA-RGS may be offered to suitable patients independently of prior PSA development and EAU BCR risk group after radical prostatectomy.

**Abstract:**

Objective: To assess the influence of biochemical recurrence (BCR) risk groups and PSA kinetics on the outcomes of radioguided surgery against prostate-specific membrane antigen (PSMA-RGS). Currently, neither BCR risk group nor PSA doubling time (PSA-DT), or PSA velocity (PSA-V) are actively assigned or relevant for counseling prior to PSMA-RGS. Methods: We retrospectively analyzed PSMA-RGS cases for oligorecurrent prostate cancer between 2014 and 2023. BCR risk groups, PSA-DT, and PSA-V were analyzed as predictors for complete biochemical response (cBR, PSA < 0.2 ng/mL), BCR-free, and therapy-free survival (BCRFS, TFS). Results: Of 374 included patients, only 21/374 (6%) and 201/374 (54%) were classified as low- and high-risk BCR (no group assignment possible in 152/374, 41%). A total of 13/21 (62%) patients with low- and 120/201 (60%) with high-risk BCR achieved cBR (*p* = 1.0). BCR classification was no predictor for BCRFS (HR:1.61, CI: 0.70–3.71, *p* = 0.3) or subsequent TFS (HR:1.07, CI: 0.46–2.47, *p* = 0.9). A total of 47/76 (62%) patients with PSA-DT ≤ 6 mo and 50/84 (60%) with PSA-DT > 6 mo achieved cBR (*p* = 0.4). PSA-DT was not associated with cBR (OR: 0.99, CI: 0.95–1.03, *p* = 0.5), BCRFS (HR: 1.00, CI: 0.97–1.03, *p* = 0.9), or TFS (HR: 1.02, CI: 0.99–1.04, *p* = 0.2). Consistent negative findings were recorded for PSA-V. Conclusions: The BCR risk groups and PSA kinetics do not predict the oncological success of PSMA-RGS performed at low absolute PSA values. Indolent low-risk BCR is rarely treated by PSMA-RGS.

## 1. Introduction

Patients with biochemical recurrence (BCR) after radical prostatectomy (RP) are often submitted to prostate-specific membrane antigen (PSMA) positron emission tomography (PET)/computed tomography (CT) for treatment planning. According to EAU guidelines, patients with a rising PSA of ≥0.2 ng/mL after RP should undergo PSMA-PET imaging if the results will influence subsequent treatment decisions [1]. In the case of oligorecurrent, hormone-sensitive prostate cancer (PCA), PSMA-targeted radioguided surgery (PSMA-RGS) has been proven effective for the removal of cancerous tissue achieving consecutive PSA responses [2].However, efficient biomarkers or imaging tools are still needed to identify patients who benefit most from this experimental procedure, and corresponding studies are ongoing (e.g., BioPoP, NCT04324983). While PSMA-RGS may especially lead to a long-lasting PSA response in less aggressive diseases (i.e., low PSA prior to PSMA-RGS), the treatment of indolent BCR may improve outcomes only marginally and represent overtreatment. No clear recommendation for salvage lymph node dissection or PSMA-RGS is provided by any guideline currently [1,3,4], and assignment to this treatment relies mainly on PSA values, imaging results (localization and count of PSMA-avid lesions), as well as patient and urologist preferences. As a relatively new procedure, clinically meaningful benefits in overall and cancer-specific survival cannot be evaluated yet.

From insights of PSA-only recurrence in the era before the widespread implementation of PSMA-PET imaging, we know about the prognostic implications of the prostate-specific antigen doubling time (PSA-DT) as a measure for disease activity at recurrence. For instance, in patients with an ISUP score ≤ 3, though not for those with higher scores, PSA-DT separated the risk of subsequent metastasis [5,6]. In other studies, PSA-DT and absolute PSA at BCR predicted metastasis-free survival in BCR after RP [7]. Based on a systematic review [8], the European Association of Urology (EAU) proposed composite risk groups for BCR after RP according to the pathologic Gleason grade group (GGG) at RP and PSA-DT [1]. In contrast to the RGS procedure, the EAU BCR risk groups were validated as an independent predictor for metastatic progression and cancer-specific mortality [9]. Furthermore, their predictive power concerning metastatic progression was recently confirmed in a multicenter trial for PSMA-PET imaging [10]. Also, EAU BCR risk stratification has been reported as a valuable decision aid for treatment planning. In detail, men with high-risk BCR showed improved survival by early salvage radiotherapy, while surveillance might be more suitable for low-risk BCR patients [11].

Although PSA-DT alone has not been assessed for PSMA-RGS so far, the GGG at RP has not yielded significant predictor status in PSMA-RGS previously [2]. In a retrospective subgroup analysis of metastasis-directed therapy (MDT) for oligometastatic PCA recurrence, shorter PSA-DT was linked to a reduced time to progression-free survival after MDT [12]. However, only 6/31 patients in this study received a surgical MDT, all procedures were without PSMA-PET guidance, and baseline characteristics were much more adverse than current PSMA-RGS cohorts.

Therefore, we evaluated whether EAU BCR risk groups as a composite and PSA Kinetics independently may aid in the selection of patients for PSMA-RGS.

## 2. Materials and Methods

### 2.1. Patient Cohort

Within the prospectively collected clinical database of two tertiary care centers, we identified 524 consecutive PSMA-RGS procedures between 2014 and 2023. In total, we excluded 150 cases due to: repeat PSMA-RGS procedures (68), (neo-)adjuvant androgen deprivation therapy prior to RGS (23), and inclusion in the prospective ProSTone trial (41; NCT04271579). A detailed enlisting of excluded cases is presented as a modified Consort flow diagram (Figure 1). For the final analysis, we included 374 patients who received primary PSMA-RGS for oligometastatic recurrence of hormone-sensitive prostate cancer after radical prostatectomy. Patients with or without pelvic radiation therapy after RP were included.

All patients were informed about the experimental nature of salvage surgery and provided their informed consent to the procedure, follow-up, and data collection/analysis. This retrospective analysis was approved by the institutional review boards of Hamburg, Germany (2019-PS-09; PV7316) and Munich, Germany (number 336/18 S). Questionnaires were used for the follow-up. All data were prospectively stored in an institutional database (FileMaker 11; Claris Inc., Cupertino, CA, USA).

### 2.2. Procedure of PSMA-RGS

The procedure of radioguided surgery against PSMA (PSMA-RGS) has been extensively described. Salvage lymph node dissection (SLND) or resection of the local recurrence was performed according to the standard extended pelvic template [1]. Unilateral or bilateral templates were performed according to the surgeon’s discretion and the patient’s preference. For presacral or pararectal lesions, resection of the corresponding region was conducted. In the case of retroperitoneal lesions, resection was performed according to the template from testicular cancer, with a pelvic SLND at least on the tumor-bearing side [2]. Radioguidance was achieved through in- and ex-vivo measurements using a gamma probe (Crystal Probe CXS-SG603 for open surgery or DROP-IN gamma probe for robotic surgery; Crystal Photonics, Berlin, Germany) with acoustic and numerical feedback as a response to ^99m^Tc radioactivity. All PSMA-RGS procedures were performed by experienced, high-volume surgeons. Dedicated and experienced uropathologists performed histopathological examinations.

### 2.3. PSA Doubling Time, PSA Velocity, and EAU Risk Group Assignment

For the calculation of PSA doubling time (PSA-DT), we applied all recommendations from the EAU guidelines [1]. In detail, we assessed all PSA values of a patient 12 months before the PSMA-RGS procedure or after the RP, whatever happened last. In the case of adjuvant or early salvage radiation therapy (RT), PSA values before and four weeks after RT were censored. Furthermore, PSA values ≤ 0.20 ng/mL were removed in accordance with EAU guidelines. A minimum period between PSA measurements of 4 weeks was requested to reduce statistical noise. If fewer than three PSA values were left for analysis, the PSA-DT of the patient was set to not assigned (NA). From the remaining PSA values, a log-linear regression model (natural logarithm of 2) for the PSA over time was generated as common practice for PSA-DT [5,13]. The PSA-DT was then assigned as the slope from the regression model. Furthermore, for global trend analysis and checking of gross violations, the investigators graphically displayed and manually reviewed the PSA development of each individual patient, including the calculated slope.

For patients with relatively constant PSA over a long time or undulating PSA, PSA-DT calculation becomes statistically more unstable [5]. Therefore, the PSA-DT was set to NA for patients with a PSA-DT of more than 5 years. This includes patients with a slope of 0 (i.e., constant PSA over time) or negative values (i.e., undulating or decreasing PSA after initial increase).

PSA-DT calculation according to the abovementioned EAU guideline recommendation has many prerequisites. Therefore, we calculated the PSA velocity (PSA-V) as the simple difference between the last PSA prior to PSMA-RGS and the first reported PSA after RP (≥0.01 ng/mL) divided by days. In the case of salvage/adjuvant radiation, the same PSA value censoring was applied as mentioned above for PSA-DT.

Accordingly, to EAU recommendation [1], patients with pathological ISUP grade ≤ 3 at RP and PSA-DT > 1 year were assigned to “low-risk BCR”. Patients with PSA-DT ≤ 1 year or pathological ISUP grade 4–5 at RP were assigned to “high-risk BCR”. In the case of ISUP grade ≤ 3 but no assigned PSA-DT, no risk group was assigned (NA).

### 2.4. Outcomes of Interest

This study’s main interest was the influence of EAU BCR risk groups and PSA kinetics on the oncological outcome of PSMA-RGS. In accordance with prior research [2,14,15,16,17,18], we assessed, therefore, complete biochemical response (cBR, defined as PSA < 0.2 ng/mL, 2–16 weeks after PSMA-RGS), biochemical recurrence-free survival (BCRFS, defined as PSA < 0.2 ng/mL without further treatment after PSMA-RGS) and therapy-free survival (TFS, defined as survival without additional therapy after PSMA-RGS) as endpoints. Patients were censored on the date of last evidence of freedom from BCR or further treatment.

### 2.5. Statistical Analyses

Descriptive statistics included frequencies and proportions for categorical variables. The medians and interquartile ranges (IQR) were reported for continuously coded variables. The statistical significance of differences in medians and proportions was evaluated using the Kruskal–Wallis and chi-square tests. For comparison of contingency tables, the Fisher’s Exact Test was applied. We assessed suspected parametric correlation of two sets of variables, i.e., PSA prior to RGS and PSA-DT or PSA-V, using Pearson correlation analysis. Univariable logistic regression models analyzed the relationship between cBR and EAU BCR risk group, PSA-DT, and PSA-V. Waterfall plots visualized the percentage of PSA change before and after PSMA-RGS within different BCR risk groups (low-risk BCR, high-risk BCR, NA) and PSA-DT groups (≤6 months, ≥6 months, NA). We chose 6 months for stratification as half of the proposed 12 months in the EAU risk classification. We verified the results with different cut-offs (3 months, 10 months, 12 months) with the same results and in accordance with the Cox Regression analysis. For survival data, Cox regression models were used to investigate the association between oncological outcomes (BRFS, TFS) and EAU BCR risk group, PSA-DT, and PSA-V. For Cox Regression, the proportional hazard assumption was tested and visualized by smoothing an appropriate residual plot as described by Grambsch et al. (part of the “survival” package in R) [19]. Kaplan–Meier plots graphically depicted the BCRFS and TFS after PSMA-RGS stratified by EAU BCR risk groups (low-risk BCR, high-risk BCR) or a PSA-DT (≤6 months, >6 months). The Log-rank method was used for the comparison of survival curves. All tests were two-sided, with the significance level pre-set at *p* < 0.05. The R software environment for statistical computing and graphics (R version 4.3.1 (16 June 2023), running under macOS Ventura 13.5) was used for all statistical analyses. The survival package (version: 3.5-5) was applied for survival analysis and Cox regression.

## 3. Results

### 3.1. Patients’ Baseline Characteristics

Overall, 374 patients were included in this analysis (Figure 1). 21/374 (5.6%) and 201/374 (53.7%) were classified as EAU low-risk or high-risk BCR, respectively. In 152/374 (40.6%) patients, risk groups could not be assigned. The median PSA [IQR] prior to RP was 6.7 ng/mL [4.3, 9.4] in the low-risk and 9.0 ng/mL [5.8, 16.0] in the high-risk BCR group (*p* < 0.05, Table 1). Per definition, the Gleason grade group (GGG) showed adverse characteristics in the high-risk group. However, GGG at RP was ≤3 in 93/201 high-risk patients; therefore, group assignment was solely performed on PSA-DT in these cases. In contrast, other baseline characteristics (age at RP, pT stage at RP, pN stage at RP, margin status at RP, RT after RP) were not statistically different between the groups (*p* > 0.05). While not statistically significant, pN0 disease, less adverse pT stage, and negative surgical margin at RP were more distinct in low-risk BCR than in the high-risk group and all patients (Table 1). A positive surgical margin was seen half as often in low-risk BCR (2/21, 10%) as in high-risk BCR (47/201, 24%) or in the entire patient cohort (80/374, 21%). Interestingly, the rate of RT after RP was evenly distributed among all groups (*p* = 1.0).

### 3.2. Patients’ Characteristics at PSMA-RGS

At PSMA-RGS, the median [IQR] age was 66 yr [61.0, 71.0] with a median [IQR] PSA value of 0.8 ng/mL [0.4, 1.7] before PSMA-RGS with no statistically relevant difference between both groups (*p* > 0.05) (Table 2). Median [IQR] time between RP and PSMA-RGS was 40.6 mo [22.6, 62.9] for low-risk and 49.6 mo [23.1, 90.7] for high-risk BCR (*p* = 0.47). In PSMA-PET imaging prior to surgery, 240 (65%), 91 (24%), 34 (9%), patients showed one, two, three or more PSMA-avid lesions, respectively. The median [IQR] number of PSMA-PET avid lesions was 1.0 (1.0, 2.0) with no difference between BCR groups (*p* = 0.7). Of these, 299 patients harbored pelvic-only disease in PSMA-PET. Per definition, median PSA-DT was higher in the high-risk group (5.5 mo [3.8, 8.0] vs. 19.2 mo [14.9, 26.5], *p* < 0.001). Interestingly, in high-risk BCR, only 31/201 patients fulfilled both inclusion criteria (GGG ≥ 4 and PSA-DT ≤ 12 mo).

### 3.3. Oncological Outcomes: cBR, BCRFS, and TFS

Follow-up data were available for 365/374 (98%) patients with a median [IQR] follow-up of 35 months [15, 53]. Within all patients, complete biochemical response (cBR, defined as post-operative PSA < 0.2 ng/mL) was seen in 207/374 (55%) patients. During follow-up, 193/374 (52%) patients experienced BCR and 130/374 (35%) patients received further therapy.

Only PSA prior to PSMA-RGS predicted cBR (OR: 0.82, 95% CI: 0.72–0.92, *p* < 0.01). Other baseline characteristics, including pT/pN Stage at RP, GGG at RP, time RP to RGS, age at PSMA-RGS, adjuvant RT after RP, number of lesions, and localization of PSMA-avid lesions, did not achieve significant predictor status for cBR in univariable logistic regression analysis (*p* > 0.05). For instance, GGG at RP (≤3 vs. ≥4, according to EAU BCR classification) did not predict cBR (OR: 1.01, 95% CI: 0.83–1.22, *p* = 0.35). The number of PET-avid lesions did not achieve significant predictor status (OR: 1.34, 95% CI: 0.98–1.87, *p* = 0.08).

The Kaplan–Meier estimates for median [95% CI] biochemical recurrence- and therapy-free survival were 19.4 mo [13.3, 34.2] and 48.6 mo [43.4, not reached]. In Cox regression, the same characteristics as mentioned above were analyzed. In accordance with the results for cBR, only PSA prior to RGS emerged as a significant predictor for BCRFS (HR: 1.06, 95% CI: 1.01–1.11, *p* < 0.05) and TFS (HR: 1.07, 95% CI: 1.01–1.13, *p* < 0.05).

#### 3.3.1. Influence of EAU Risk Stratification

Immediately after PSMA-RGS, 13/21 (62%) patients with low-risk and 120/201 (60%) with high-risk BCR achieved complete biochemical response (cBR, defined as post-operative PSA < 0.2 ng/mL) with no statistically relevant difference between these two groups (*p* = 1.0). Graphically, a solid PSA response after PSMA-RGS was seen among both groups (Figure 2). High-risk BCR was not associated with cBR in univariable logistic regression analysis (OR: 0.94, 95% CI: 0.32–2.50, *p* = 0.9) (Table 3).

In Cox regression, BCR risk groups did not statistically significantly predict BCRFS (HR: 1.61, 95% CI 0.70–3.71, *p* = 0.3) (Table 3). The Kaplan–Meier estimate for the median [95% CI] BCRFS was 32.8 mo [21.8, 57] for high-risk BCR patients (Figure 3).

In Cox regression, BCR risk classification did not statistically significantly predict TFS (HR: 1.07, 95% CI: 0.46–2.47, *p* = 0.9) (Table 3). The Kaplan–Meier estimate for the median [95% CI] TFS was 52.7 mo [46.8, not reached] for high-risk BCR (Figure 3).

While in 365/374 patients follow-up information was available, the Kaplan–Meier estimates only present patients with given group assignments.

For low-risk BCR, the calculation of the median BCRFS and TFS was not possible due to the small sample size.

#### 3.3.2. Influence of PSA Kinetics

The median [IQR] PSA-DT was 6.2 mo [4.0, 9.9]. The distribution curve had a bell shape (Appendix A). PSA-DT could be calculated in 160/374 (43%) patients. Of these, 76/160 and 84/160 presented with a PSA-DT ≤ 6 mo or > 6 mo. A total of 47/76 (62%) patients with PSA-DT ≤ 6 mo and 50/84 (60%) with PSA-DT > 6 mo achieved cBR (*p* = 0.4). Graphically, a solid PSA response after PSMA-RGS was seen among both PSA-DT groups (Figure 2). PSA-DT did not correlate with PSA prior to RGS in a statistically relevant manner (Pearson correlation coefficient: −0.09, *p* = 0.3).

PSA-DT (as a continuous variable) was not associated with cBR in univariable logistic regression analysis (OR: 0.99, 95% CI: 0.95–1.03, *p* = 0.5) (Table 3).

In Cox regression, PSA-DT did not statistically significantly predict BCR-free survival (HR: 1.00, 95% CI: 0.97–1.03, *p* = 0.9) (Table 3). The Kaplan–Meier estimates for median [95% CI] BCRFS was 44.6 [14.6., not reached] for PSA-DT ≤ 6 mo and 47.4 mo [28.2, not reached] for PSA-DT > 6 mo (Figure 3).

In Cox regression, PSA-DT did not statistically significantly predict TFS (HR: 1.02, 95% CI: 0.99–1.04, *p* = 0.2) (Table 3). The Kaplan–Meier estimates for the median [95% CI] TFS were 48.0 [35.5, not reached] for PSA-DT ≤ 6 mo and 52.8 mo [48.0, not reached] for PSA-DT > 6 mo (Figure 3).

PSA-V could be calculated in 302/374 (81%) patients. The median [IQR] PSA-V was 0.46 ng/mL/year [0.18, 1.29]. In summary, PSA-V did not achieve significant predictor status for cBR, BCRFS, or TFS (Table 3). PSA-V did not correlate with PSA prior RGS (Pearson correlation coefficient: 0.06, *p* = 0.3).

## 4. Discussion

For oligorecurrent prostate cancer after local treatment, MDT has been proposed to delay systematic treatment [1,20,21]. However, patient selection remains difficult, and overtreatment of indolent BCR is a common concern. To the best of our knowledge, the influence of EAU BCR risk groups and PSA kinetics has not been assessed in the context of PSMA-RGS before. Therefore, our analyses demonstrated several important findings.

First, only a few patients who present with oligorecurrent PCA deemed suitable for PSMA-RGS have low-risk BCR (21/374 (all patients), 6%; or 21/222 (patients with risk group assignment), 9%). EAU risk stratification for PSA-only recurrence might be applied for PSMA-RGS candidates despite harboring evident (nodal) metastases in PSMA-PET/CT because these metastases are often not seen in conventional imaging. Most landmark studies concerning the first-line treatment of metastatic prostate cancer rely on conventional imaging for patient inclusion [22,23,24,25,26,27,28]. Therefore, PSMA-RGS candidates may be located between (indolent, equal to low-risk) PSA-only recurrence and early metastatic disease. Accordingly, PSMA-RGS patients represent mostly high-risk BCR by conventional measures. The minor rate of low-risk BCR within our cohort disproves overtreatment of indolent PSA-only recurrence by PSMA-RGS as a common concern. PSA monitoring should only be offered to EAU low-risk BCR patients [1]. Most PSMA-RGS patients would be candidates for hormonal therapy in combination with local radiotherapy if not already performed [1]. This finding is not surprising, because high-risk BCR is an independent predictor for metastatic progression [9]. In contrast, in the validation cohort of Tilki et al., all patients with BCR after RP were included, and EAU BCR risk groups were evenly distributed (low risk: 510/1040, high risk: 530/1040, *p* = 0.5) [9]. Whilst PSA-DT after RP and high GGG at RP offer guidance in an overall BCR population for prognosis [8] and treatment planning like salvage RT [11], the added predictive value of EAU BCR risk stratification in this pre-selected cohort is minimal. In detail, early oligorecurrent PCA in PSMA-PET/CT may represent high-risk PSA-only recurrence in conventional imaging.

Second, PSA kinetics are also not predictive in PSMA-RGS patients. Cancer is a growth process. Consequently, the growth function’s slope measured by PSA as a surrogate for tumor volume expresses the aggressiveness of cancer recurrence. GGG at RP has not significantly predicted oncological effectiveness at PSMA-RGS in prior analysis [2]. The composite definition of GGG and PSA-DT for BCR risk stratification was mainly chosen because PSA-DT firmly separated the risk of subsequent metastasis in patients with GGG ≤ 3 [5,6,7,8]. Besides these theoretical considerations, neither PSA-DT nor PSA-V independently achieved significant predictor status on any parameter for oncologic outcomes (cBR, BCRFS, TFS) in our analysis. Consistent with this, Ost et al. found no significant interaction between the effect of MDT and PSA-DT in a subgroup analysis of the prospective, randomized, multicenter phase II trial comparing surveillance vs. MDT in oligorecurrence of PCA [12]. In M0-BCR patients, ADT as systematic salvage treatment should not be offered if PSA-DT is >12 months, and deferred treatment is advisable because of its indolent nature [1]. However, only 29/374 patients (8%; 29/160 (18%) of patients with an assigned PSA-DT) had a PSA-DT > 12 months within our cohort. Interestingly, the number and localization of PET-avid lesions did not achieve overall significant predictor status within this pre-selected cohort, in contrast to prior analysis [2]. While this might be seen as an expression of the limited sample size or changes in patient selection over time, the concrete odds and hazard ratios of around one for all investigated parameters raise the question of whether results might have changed within a larger cohort. Still, we believe that PSMA-RGS should primarily be offered to patients with oligorecurrence and favorably only one lesion in accordance with our earlier analyses.

Third, PSA-DT may help to identify patients who should be screened by PSMA-PET/CT for oligorecurrence for potential PSMA-RGS or MDT in general. The perfect timing of molecular imaging with a significant impact on treatment decisions remains to be determined. In very early BCR (PSA < 0.2 ng/mL), PSMA-PET/CT is often negative [29]. Therefore, PSMA-PET/CT is currently recommended based on PSA [1]. Recently, signals for a correlation of PSA kinetics and positive PSMA-PET/CT have been seen [30,31]. A recent meta-analysis showed a pooled OR of 3.22 (95% CI: 1.17–8.88) of a positive PSMA-PET in patients with PSA-DT ≤ 6 months compared to patients with PSA-DT > 6 months. However, these results did not achieve statistical significance, and high heterogeneity among included studies was found (I^2^ index: 80%) [31]. In contrast, patients with very early recurrence, i.e., one pelvic lesion at low PSA before RGS, benefit most from PSMA-RGS [2]. While PSA-DT may not be of considerable value in patients with known oligorecurrence, the homogenous distribution of PSA-DT within PSMA-RGS patients (median: 6.1, IQR: 4.0–9.9) may showcase which patients should be screened for oligorecurrence for potential MDT by PSMA-PET/CT. Because of the relatively indolent nature of low-risk BCR patients, this might be especially true within BCR high-risk patients (median PSA-DT: 5.5, IQR: 3.8–8.0).

However, several limitations of our study must be mentioned. First, only a small proportion of our cohort harbored low-risk BCR (21/374, 6%) and, therefore, the groups within the analysis are unbalanced. In general, the inverse-stage migration in BCR risk groups at PSMA-RGS may be explained through selection bias at referral/counseling and the potentially more rare prevalence of PSMA-avid oligorecurrence in BCR low-risk patients.

For referral, patients with low-risk BCR might be referred less frequently for potential surgery at our institution because surveillance is recommended for these patients in current guidelines [1]. While we cannot exclude subconscious selection bias, neither the BCR risk group nor PSA-DT is actively assigned during the counseling process nor requested for the referral. Roughly half of the patients in our database (177/374, 47%) do not provide sufficient PSA values to calculate PSA-DT. Even though patients ordinarily present with the pathology results from RP at counseling for potential PSMA-RGS, the GGG at RP has a relatively small influence on our recommendation because GGG at RP never reached significant predictor status in earlier analyses [2].

On the other hand, PSA at BCR [30] and the BCR risk groups [10,32] define candidates who benefit most from PSMA-PET/CT. In one study by Dong et al., for instance, the BCR risk group was independently associated with positive PSMA-PET findings (OR: 6.73, 95% CI: 2.41–18.76, *p* < 0.001) and lymph node involvement (OR: 2.38, 95% CI: 1.04–5.49, *p* < 0.05) in an all-comer population of BCR patients after RP [32].

Finally, the value of MDT in general and PSMA-RGS in detail for long-term clinical meaningful improvement of cancer-specific and overall survival has yet to be established. The reduction in an asymptomatic PSA value has no clinical value per se. While delaying subsequent treatment may show clinical value, this must be assessed cautiously, because treatment decisions are often based on PSA values. However, in unselected relapsing patients, the median actuarial time to the development of metastasis and consequent death is around 8 years and another 5 years, respectively [1,6]. Therefore, we need patience to see the data of a new procedure maturate.

To conclude, in patients who are candidates for PSMA-RGS, neither PSA kinetics nor EAU BCR risk stratification offers guidance. Overtreatment of indolent BCR by PSMA-RGS might be more limited than expected because mainly high-risk BCR patients are selected for surgery. The inherent correlation between PSMA-avid lesions and adverse baseline characteristics might limit the number of PSMA-RGS candidates even before referral/selection bias occurs. Further research on BCR risk stratification incorporating molecular imaging and MDT is necessary.

## 5. Conclusions

EAU BCR risk stratification and PSA-kinetics, i.e., PSA doubling time and velocity, are no significant predictors for the oncological success of PSMA-RGS performed at low absolute PSA values. In this selected cohort of mainly patients with one lesion, only PSA prior to PSMA-RGS was a significant predictor of improved cancer-specific outcomes. Therefore, PSMA-RGS may be offered to suitable patients independently of prior PSA development and Gleason Grade at RP.

## Figures and Tables

**Figure 1 cancers-15-05008-f001:**
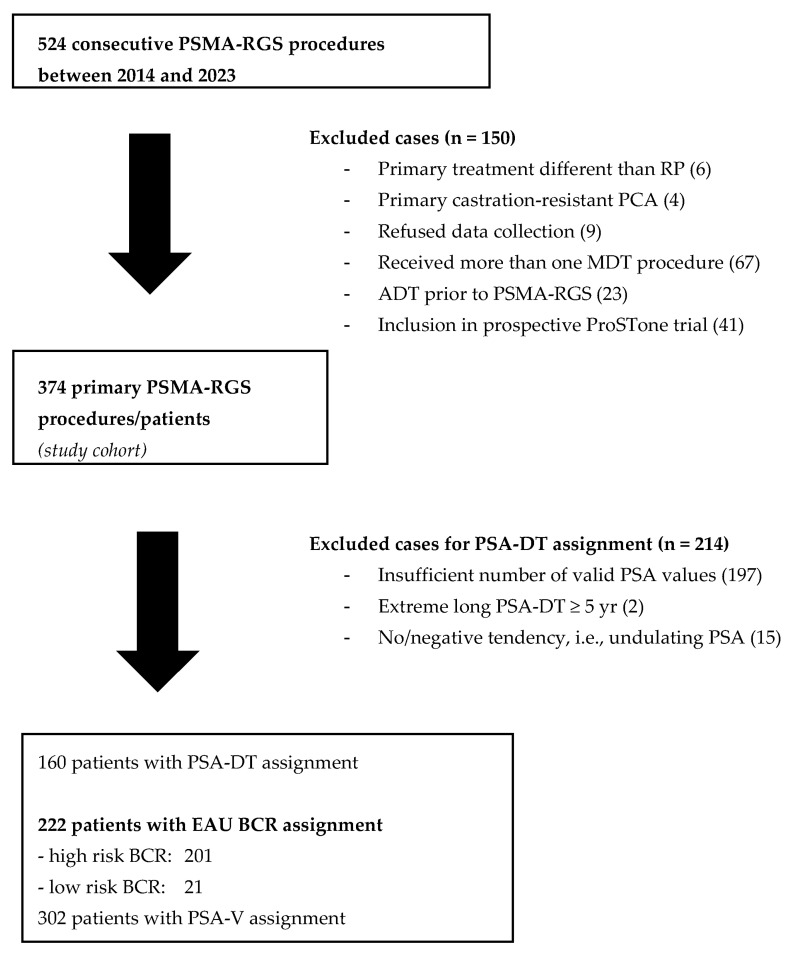
Modified Consort diagram of the study cohort treated with PSMA-RGS between 2014 and 2023 in two tertiary care centers. RP = radical prostatectomy; PCA = prostate cancer; PSMA-RGS = radioguided surgery against PSMA; MDT = metastasis-directed therapy such as PSMA-RGS; ADT = androgen deprivation therapy; IRE = irreversible electroporation; RT = radiotherapy; PSA-DT = PSA doubling time; PSA-V = PSA velocity.

**Figure 2 cancers-15-05008-f002:**
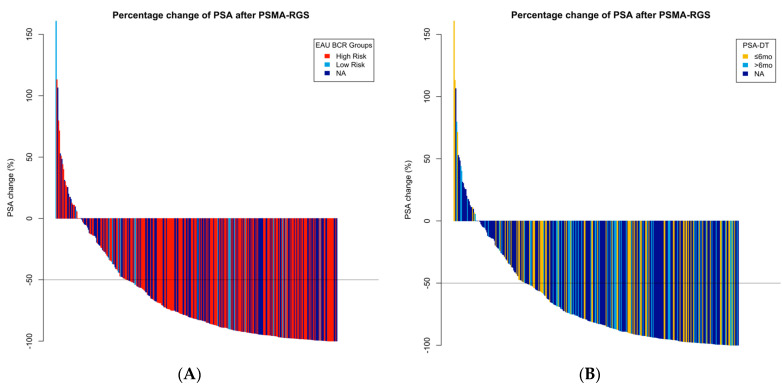
Waterfall plots graphically depicting the percentage of PSA change from before and after PSMA-RGS, color-coded according to EAU BCR risk groups (**A**); high-risk BCR, low-risk BCR, Not Assigned) and PSA-DT (**B**) ≤6 months, >6 months, Not Assigned). PSA-DT = PSA doubling time; PSA-V = PSA velocity; mo = months.

**Figure 3 cancers-15-05008-f003:**
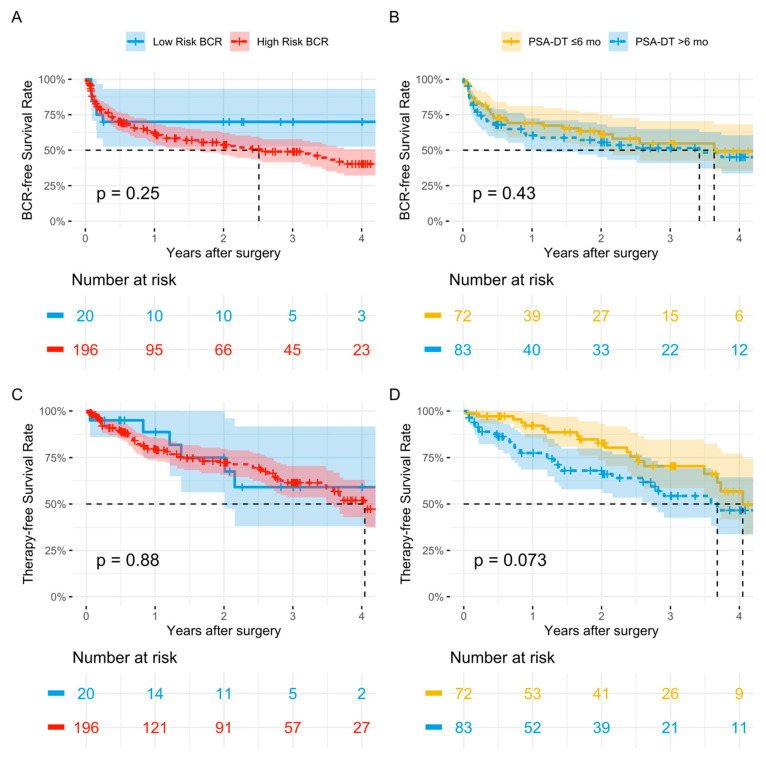
Kaplan–Meier estimates depicting biochemical recurrence- and therapy-free Survival Rates (BCRFS and TFS) in 364 patients treated with PSMA-RGS between 2014 and 2023 in two tertiary care centers. (**A**) BCRFS stratified according to EAU BCR risk groups (Low-Risk BCR, High-Risk BCR). (**B**) BCRFS stratified according to PSA-DT (≤6 months, >6 months). (**C**) TFS stratified according to EAU BCR risk groups (Low-Risk BCR, High-Risk BCR). (**D**) TFS stratified according to PSA-DT (≤6 months, >6 months). PSMA-RGS = radioguided surgery against PSMA.

**Table 1 cancers-15-05008-t001:** Baseline characteristics of 374 patients treated with PSMA-RGS between 2014 and 2023 in two centers, stratified according to the EAU BCR risk groups (low-risk, high-risk, and all patients (including patients with no assigned risk group)).

Characteristic	Low-Risk BCR (*n* = 21)	High-Risk BCR (*n* = 201)	All Patients (*n* = 374)	*p*
Age at RP (yr), median (IQR)	63.0 (54.0, 68.0)	60.0 (54.0, 65.0)	60.0 (55.0, 65.0)	0.4
PSA at RP (ng/mL), median (IQR)	6.7 (4.3, 9.4)	9.0 (5.8, 16.0)	8.5 (5.7, 14.8)	0.03
GGG at RP, n (%)				
1	0 (0)	10 (5)	27 (7)	< 0.001
2	5 (24)	41 (20)	93 (25)	
3	16 (76)	42 (21)	132 (35)	
4	0 (0)	40 (20)	40 (11)	
5	0 (0)	61 (30)	61 (16)	
NA	0 (0)	7 (4)	21 (6)	
pT Stage at RP, n (%)				
pT2	8 (38.1)	65 (32.3)	141 (37.7)	0.9
pT3a	7 (33.3)	68 (33.8)	111 (29.7)	
pT3b	6 (28.6)	65 (32.3)	106 (28.3)	
NA	0 (0.0)	3 (1.5)	16 (4.3)	
pN stage at RP, n (%)				
pN0	18 (85.7)	151 (75.1)	275 (73.5)	0.5
pN1	3 (14.3)	38 (18.9)	64 (17.1)	
pNx/NA	0 (0.0)	12 (5.6)	35 (9.4)	
Surgical Margin at RP, n (%)				
R0	19 (90.5)	142 (73.6)	262 (70.1)	0.2
R1	2 (9.5)	47 (24.4)	80 (21.4)	
Rx/NA	0 (0.0)	12 (6.0)	32 (8.6)	
RT after RP, n (%)	12 (57)	116 (58)	215 (58))	1.0

BCR = Biochemical Recurrence; GGG = Gleason grade group; PSA = prostate-specific antigen; RP = radical prostatectomy; RT = radiotherapy; yr = years; pT and pN stage according to the American Joint Committee on Cancer TNM manual.

**Table 2 cancers-15-05008-t002:** Pre-operative characteristics of 374 patients treated with PSMA-RGS between 2014 and 2023 in two centers, subdivided by EAU BCR risk groups (low-risk, high-risk, and all patients (including patients with no assigned risk group)).

Characteristic	Low-Risk BCR (*n* = 21)	High-Risk BCR (*n* = 201)	All Patients (*n* = 374)	*p*
Age at PSMA-RGS (yr), median (IQR)	66.0 (57.0, 71.0)	65.0 (61.0, 70.0)	66.0 (61.0, 71.0)	0.8
Time between RP and PSMA-RGS (mo), median (IQR)	40.6 (22.6, 62.9)	49.6 (23.1, 90.7)	50.4 (24.4, 91.8)	0.5
PSA prior PSMA-RGS (ng/mL), median (IQR)	0.5 (0.3, 1.2)	0.8 (0.4, 1.4)	0.8 (0.4, 1.7)	0.2
PSA doubling time (months), median (IQR)	19.2 (14.9, 26.5)	5.5 (3.8, 8.0)	6.1 (4.0, 9.9)	<0.001
PSA velocity (ng/mL/month), median (IQR)	0.3 (0.1, 0.5)	0.6 (0.2, 1.7)	0.5 (0.2, 1.3)	0.02
No. of PSMA-PET avid lesions				
1—n (%)	13 (61.9)	132 (65.7)	240 (64.2)	0.6
2—n (%)	6 (28.6)	45 (22.4)	91 (24.3)	
≥3—n (%)	2 (9.5)	20 (10.0)	34 (9.1)	
NA	0	4 (2.0)	9 (2.4)	
PSMA PET localization				
Pelvic only, n (%)	16 (76.2)	161 (82.6)	299 (82.6)	0.6
Pelvic + retro, n (%)	3 (14.3)	15 (7.7)	31 (8.6)	
Retroperitoneal, n (%)	2 (9.5)	19 (9.7)	32 (8.8)	

BCR = biochemical recurrence; cBR = complete biochemical response, defined as post-operative PSA <0.2 ng/mL; PSA = prostate-specific antigen; PET = positron emission tomography; PSMA = prostate-specific membrane antigen; PSMA-RGS = radioguided surgery against PSM; yr = years; mo = months.

**Table 3 cancers-15-05008-t003:** Univariable Logistic (for cBR) and Cox (for BCRFS and TFS) regression models assessing the influence of BCR risk group (low- vs. high-risk BCR), PSA-DT (continuously) and PSA-V (continuously) on oncological outcomes.

	cBR (Logistic Regression)				BCRFS (Cox Regression)				TFS (Cox Regression)			
	OR	95% CI		*p*	HR	95% CI		*p*	HR	95% CI		*p*
EAU BCR												
-low risk	Ref.											
-high risk	0.94	0.32	2.50	0.9	1.61	0.70	3.71	0.3	1.07	0.46	2.47	0.9
PSA-DT, in mo (cont.)	0.99	0.95	1.03	0.5	1.00	0.97	1.03	0.9	1.02	0.99	1.04	0.2
PSA-V, in ng/mL/yr (cont.)	1.02	0.98	1.09	0.5	1.01	0.99	1.03	0.4	1.01	0.99	1.03	0.4

Cont. = continuously coded; OR = Odds Ratio, HR = Hazard Ratio, 9% CI = 95% Confidence Interval, PSA-DT = PSA doubling time, PSA-V = PSA velocity.

## Data Availability

The data presented in this study are available upon request.

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
