# Peer review of "EAU Biochemical Recurrence Risk Classification and PSA Kinetics Have No Value for Patient Selection in PSMA-Radioguided Surgery (PSMA-RGS) for Oligorecurrent Prostate Cancer"

_cancers, 2023, doi:10.3390/cancers15205008_

Round 1

Reviewer 1 Report

With great interest I have reviewed your paper entitled EAU Biochemical Recurrence Risk Classification and PSA Kinetics have limited value for patient selection in PSMA-radioguided Surgery (PSMA-RGS) for Oligorecurrent Prostate Cancer. In this well written paper the authors analyzed if the EAU risk stratification for biochemical recurrence (BCR) and the PSA development over time can predict the oncological success of PSMA-RGS. This is a very relevant question since patient selection for salvage therapy is a very important element in having the best impact on patients' oncological outcomes.  The manuscript is well written with a nice presentation of the data.

I have the following comments / suggestions.

Title: why limited effect? In the conclusion EAU risk classification and PSA kinetics have no effect on outcome

Abstract: Mention limitation that patients were already selected based on ..... ? maybe they were already selected based on PSA kinetics?. Also mention that only 21/374 were classified as BCR low risk which is a very low number for statistical analysis. This is of course a good observation because these patient could also benefit from observation and for subsequently not included in your surgical cohort study. Also make clear that data was complete in .... number of patients. Now its 21/374 low risk and 201/374 with high risk..... what happened with the 152 patients?

Introduction: Include that according to the guideline patients with rising PSA (0.2) after RP should undergo PSMA PET (weak recommendation)

Methods:

- Could you provide some background information on statistical power calculation comparing outcomes n=21 low risk versus 201 high risk

- We know from previous work on this cohort that PSMA PET location and number of pos PSMA PET lesions have significant impact on the outcomes. You might consider to make a sub analyses only including patients with 1 PSMA PET pos lesion limited to the pelvis

- Could you briefly explain why reporting on PSAdt and PSA velocity. Use reference to explain that outcomes can be different especially in low PSA values.

Conclusions

I agree that based on your findings PSA kinetics have no impact on oncological outcomes after RGS. However based on previous publication (e.g. PSA < 1, one PSMA PET positive lesion) you could mention clinical parameters that have impact just to guide the readers who are interested in selecting patients for PSMA RGS.

With great interest I have reviewed your paper entitled EAU Biochemical Recurrence Risk Classification and PSA Kinetics have limited value for patient selection in PSMA-radioguided Surgery (PSMA-RGS) for Oligorecurrent Prostate Cancer. In this well written paper the authors analyzed if the EAU risk stratification for biochemical recurrence (BCR) and the PSA development over time can predict the oncological success of PSMA-RGS. This is a very relevant question since patient selection for salvage therapy is a very important element in having the best impact on patients' oncological outcomes.  The manuscript is well written with a nice presentation of the data.

I have the following comments / suggestions.

Title: why limited effect? In the conclusion EAU risk classification and PSA kinetics have no effect on outcome

Abstract: Mention limitation that patients were already selected based on ..... ? maybe they were already selected based on PSA kinetics?. Also mention that only 21/374 were classified as BCR low risk which is a very low number for statistical analysis. This is of course a good observation because these patient could also benefit from observation and for subsequently not included in your surgical cohort study. Also make clear that data was complete in .... number of patients. Now its 21/374 low risk and 201/374 with high risk..... what happened with the 152 patients?

Introduction: Include that according to the guideline patients with rising PSA (0.2) after RP should undergo PSMA PET (weak recommendation)

Methods:

- Could you provide some background information on statistical power calculation comparing outcomes n=21 low risk versus 201 high risk

- We know from previous work on this cohort that PSMA PET location and number of pos PSMA PET lesions have significant impact on the outcomes. You might consider to make a sub analyses only including patients with 1 PSMA PET pos lesion limited to the pelvis

- Could you briefly explain why reporting on PSAdt and PSA velocity. Use reference to explain that outcomes can be different especially in low PSA values.

Conclusions

I agree that based on your findings PSA kinetics have no impact on oncological outcomes after RGS. However based on previous publication (e.g. PSA < 1, one PSMA PET positive lesion) you could mention clinical parameters that have impact just to guide the readers who are interested in selecting patients for PSMA RGS.

Author Response

To Ms. Zooey Zhang

Assistant Editor, Cancers

Hamburg, 30th of September 2023

Dear Ms. Zooey Zhang,

On behalf of all authors, we thank the reviewers for their important and constructive comments, which improved the overall quality of the manuscript. We have reviewed and corrected the manuscript (cancers-2631232) entitled: "EAU Biochemical Recurrence Risk Classification and PSA Kinetics have no limited value for patient selection in PSMA-radioguided Surgery (PSMA-RGS) for Oligorecurrent Prostate Cancer", according to the reviewers' suggestions, comments, and underlying thoughts. 

The revised manuscript provides a detailed outline of the changes in yellow. We hope that the Editorial Board will find the revised version of the manuscript satisfactory for publication.

Yours sincerely,

Prof. Dr. Tobias Maurer, on behalf of all authors

Martini-Klinik Prostate Cancer Center and Department of Urology

University Medical Center Hamburg-Eppendorf

Martinistr. 52, 20246 Hamburg, Germany

Mail:               [email protected]

We thank the reviewers for their valuable comments. A point-by-point response is provided below. In our responses to the reviewers, we have highlighted the changes in the text by marking the adapted passages in yellow in the revised manuscript. To meet the word count, we have streamlined the abstract after including reviewer feedback.

Comments to Authors:

Reviewer #1:           […] Title: why limited effect? In the conclusion EAU risk classification and PSA kinetics have no effect on outcome […]

Authors` response: 

We thank the reviewer for this important comment. Indeed, the EAU BCR risk groups and PSA kinetics did not aid in patient selection for PSMA-RGS in our cohort. Initially, we had chosen "limited value" in the title because of the strong selection bias and underrepresentation of EAU BCR low risk patients. After internal discussion and to improve the manuscript's clarity, we have changed the title accordingly:

"EAU Biochemical Recurrence Risk Classification and PSA Kinetics have no value for patient selection in PSMA-radioguided Surgery (PSMA-RGS) for Oligorecurrent Prostate Cancer"

Reviewer #1:           Abstract: […] Mention limitation that patients were already selected based on...? maybe they were already selected based on PSA kinetics?[…]

Authors` response: 

We thank the reviewer for raising awareness regarding selecting patients for PSMA-RGS. We completely agree with the strong implications of selection bias within this cohort at many levels (frequency of PSA testing, early referral for PSMA-PET/CT, referral for surgery, and counseling prior to surgery). At least for counseling prior to a possible PSMA-RGS at our institution, neither the BCR risk group nor PSA-DT or PSA-V was actively assigned. In fact, we often have only the PSA value prior to radical prostatectomy and the current PSA value at referral. Patient selection was mainly performed based on the number and location of the PSMA-avid lesion and general considerations such as age and comorbidities.

In the Abstract, we added the following:

[…] Currently, neither BCR risk group nor PSA doubling time(PSA-DT), or PSA velocity(PSA-V) are actively assigned or relevant for counseling prior PSMA-RGS. […]

Reviewer #1:           Abstract: […] Also mention that only 21/374 were classified as BCR low risk which is a very low number for statistical analysis. This is of course a good observation because these patient could also benefit from observation and for subsequently not included in your surgical cohort study. […]

Authors` response: 

We thank the reviewer for this comment and completely agree. One major fear of performing (any) metastasis-directed therapy at low PSA values is the overtreatment of indolent PSA recurrence.Therefore, the minor rate of EAU low risk BCR (6%) shows that we mainly targeted patients with aggressive disease, although BCR risk groups were not calculated before. To emphasize, we have added the percentage of each group in the results and stressed the low prevalence of low risk BCR in the conclusion. 

In the Abstract, we added the following:

[…]Of 374 included patients, only 21/374(6%) and 201/374(54%) were classified as low and high risk BCR[…]  Indolent low risk BCR is rarely treated by PSMA-RGS. […]

Reviewer #1:           Abstract: […]Also make clear that data was complete in .... number of patients. Now its 21/374 low risk and 201/374 with high risk..... what happened with the 152 patients? […]

Authors` response: 

We thank the reviewer for raising awareness concerning the clear understandability of the presented data. In the "missing" 152 patients, no group assignment was possible.  The EAU guidelines are very strict concerning the PSA-DT calculation and BCR risk group assignment, as described in the Methods section. To accommodate the high "not assigned"-rates in risk groups and PSA-DT, we further analyzed PSA-V independently (PSA-V not assigned in 72/374 patients, 19%).

In the Abstract, we added the following:

[…]Of 374 included patients, only 21/374(6%) and 201/374(54%) were classified as low and high risk BCR (no group assignment possible in 152/374, 41%).[…]

Reviewer #1:           Introduction: Include that according to the guideline patients with rising PSA (0.2) after RP should undergo PSMA PET (weak recommendation).

Authors` response: 

We agree with the reviewer regarding the EAU guideline recommendations and have added them accordingly to the Introduction.

In the Introduction, we have added the following:

[…]According to EAU guidelines, patients with a rising PSA of ≥ 0.2 ng/ml after RP should undergo PSMA-PET imaging if the results will influence subsequent treatment decisions […]

Reviewer #1:           Methods: […]  Could you provide some background information on statistical power calculation comparing outcomes n=21 low risk versus 201 high risk […]

Authors` response: 

We thank the reviewer for their comment. As usual in observational research (this trial was retrospective), we did not perform any sample size or a priori power calculation before choosing the cohort. The cohort size was chosen based on "convenience"; i.e., all patients within our institutional databases were included who were not excluded for several reasons (see Figure 1: Modified Consort diagram).

However, the underlying thoughts of the reviewer were valid. For instance, we calculated the post hocpower for complete biochemical response between low and high risk BCR of only 11%.  Because post hoc power analyses are typically only calculated for nonsignificant/negative trials (like ours), this results automatically in low power values and may mislead the reader to the (false) conclusion that the trial just had an inadequate sample size/power (see: Althouse A. D. (2021). Post Hoc Power: Not Empowering, Just Misleading. The Journal of surgical research, 259, A3–A6.). Therefore, we prefer to report the risks with corresponding 95% confidence intervals and p-values to determine the statistical power. In fact, the sample size at the given enrollment ratio of low vs. high risk BCR (roughly 1:10, without NA patients!) and a potential difference of 10% in cBR must exceed 2000 patients (>4000 patients with NA patients) to achieve a statistical power of 80% with an alpha of 0.05.

Reviewer #1:           Methods: […]  We know from previous work on this cohort that PSMA PET location and number of pos PSMA PET lesions have significant impact on the outcomes. You might consider to make a sub analyses only including patients with 1 PSMA PET pos lesion limited to the pelvis  […]

Authors` response: 

We entertained this idea with great interest. However, subgroup analysis of this cohort (one PSMA-avid lesion in the pelvis) did not alter the analysis results. For instance, the BCR-free survival in the above-mentioned subgroup stratified for the BCR risk group or PSA-DT group was as follows:

Reviewer #1:           Methods: […] Could you briefly explain why reporting on PSAdt and PSA velocity. Use reference to explain that outcomes can be different especially in low PSA values.[…]

Authors` response: 

We thank the reviewer for raising awareness concerning the modeling of PSA development over time, which can be quite challenging using real-world data. First, we calculated the PSA-DT based on the EAU guideline recommendations and common practice (Vickers et al., Br J Med Surg Urol. 2012).  The PSA-DT is the slope of a log-linear regression model (natural logarithm of 2) with at least three PSA values of certain properties within the last 12 months. Although the strict recommendations for PSA value censoring and PSA-DT calculation are statistically sound and make comparisons with different landmark studies of BCR risk groups possible, many patients cannot be assigned a PSA-DT in real-world data, especially in patients with low PSA values (214/374 patients (57%) with no assigned PSA-DT).

To overcome this challenge and include more patients' PSA development for analysis, many options were discussed, and a plethora of different definitions for PSA kinetics exists (for instance, in one article 22 definitions were mentioned; see: O’Brien et al. J Clin Oncol. 2009). In contrast to defining a "second" less strict PSA-DT, we chose PSA-V as the simple difference between the last PSA prior to PSMA-RGS and the first reported PSA after RP (≥0.01ng/ml) divided by days. While this is statistically not devoid of limitations and the log-linear regression model fits better for the growth pattern than a simple linear relationship, the number of patients in whom the PSA history could be elevated by this definition nearly doubled (72/374 patients (19%) with no assigned PSA-V). 

To conclude, no model perfectly fits low PSA development in a real-world cohort. Therefore, we offered a very precise and validated method (PSA-DT) and a less robust method with more patients included (PSA-V).

Reviewer #1:           Conclusions: […]I agree that based on your findings PSA kinetics have no impact on oncological outcomes after RGS. However based on previous publication (e.g. PSA < 1, one PSMA PET positive lesion) you could mention clinical parameters that have impact just to guide the readers who are interested in selecting patients for PSMA RGS.[…]

Authors` response: 

We thank the reviewer for this recommendation. In this study, only PSA prior PSMA-RGS predicted cBR (OR: 0.82, 95% CI: 0.72-0.92, p < 0.01), BCRFS (HR: 1.06, 95% CI: 1.01-1.11, p < 0.05) and TFS (HR: 1.07, 95% CI: 1.01-1.13, p < 0.05). The number of PSMA-avid lesions did not achieve a significant predictor status in this analysis, which might be explained by the changing percentage of patients with only one lesion prior to PSMA-RGS, in contrast to earlier analyses.

In the Discussion section, we have added the following:

[…]Still, we believe that PSMA-RGS should primarily offered to patients with oligo-recurrence and favorable only one lesion in accordance with our earlier analyses. […]

In the Conclusion section, we have added the following:

[…]In this selected cohort of mainly patients with one lesion, only PSA prior to PSMA-RGS was a significant predictor of improved cancer-specific outcomes. […]

Reviewer #2:            […]Maybe other covariates could be evaluated (lympho-vascular invasion, cribriform presentation etc.) also.  Considering that the majority of cases were classifies as high-risk it would have been interesting to assess only this subgroup according the different pathologic biomarkers. […]

Authors` response: 

We thank the reviewer for the insightful comment. Indeed, many histopathological characteristics and biomarkers may be useful for further analysis. A prospective biomarker study is currently ongoing at our center (BioPoP, NCT04324983), and we hope to report our results on potential biomarkers in the future. As most initial radical prostatectomies are performed in other hospitals, we, unfortunately, do not have standardized information on lymphovascular invasion and cribriform presentation for most patients and thus cannot provide these analyses. Further studies concerning the pathological information and outcomes of PSMA-RGS are needed. 

Reviewer 2 Report

The authors aimed to assess the possible role of EAU risk groups  and PSA kinentics ti predict oncological outcome after PSMA-RGS. 

The study tries to answer to contemporary clinical needs considering the increasing implementation of MDT in oligo-reccurent disease after radical prostatectomy. However the therapeutic role of MDT is still debated but the authors clarified this point explicitly. 

It is of interest that common biomarkers were analyzed. Maybe other covariates could be evaluated (lympho-vascular invasion, cribriform presentation etc.) also. 

Considering that the majority of cases were classifies as high-risk it would have been interesting to assess only this subgroup according the different pathologic biomarkers.

In general it is a well written article. It shows negative results but these are too often underreported. 

I would like to congratulate the authors for the scientific effort.

Kind regards

No major observations. 

Author Response

(The authors gave the same response as above.)
